# Digital and Sharing Economy for Sustainable Development: A Bibliometric and Systematic Review

**Osho Awli and Evan Lau \***

Faculty of Economics and Business, Universiti Malaysia Sarawak, Kota Samarahan 94300, Malaysia
\* Correspondence: lphevan@unimas.my

**Abstract:** The concept of digitalization has become a common practice for adoption and integration across the economy in recent years. The rapid prospects of a spillover from digitalization quickly became the interest of countries and researchers, especially in the interest of sustainable development based on the SDGs of the United Nations. With several subsectors of the economy surfacing as a product of the digital economy, this study intends to identify the relationship between the digital economy and sharing economy, as well as their role in sustainable development. To achieve the objectives laid out, this study employs the use of bibliometric analysis and systematic literature review (SLR) to organize and extract the contents of the selected literature. The findings show that the contemporary sharing economy is driven by the digital economy and is reliant on its digital infrastructure, whereas there are mixed findings on the role of digitalization on sustainability.

**Keywords:** digital economy; sharing economy; sustainable development; bibliometric analysis; systematic literature review

## 1. Introduction

Digitization and digitalization have become an integral, necessary change in the global economy, changing the trends of society and business in both the short and long term (Ailisto et al. 2016). As stated by Ojanperä et al. (2019), the digital revolution is a holistic one which has seeped into all aspects of the economy, systemically superseding inferior traditional systems with the transformative power of an informational revolution. The growing interest and the adoption of digitalization across all sectors in the economy has birthed digitally centric sectors of the economy, such as the digital economy and sharing economy, which call to attention the bounds and parameters of these sectors. As core interests, digitization and digitalization are seen to be the byproducts of accelerated globalization, where digitization is the conversion of information into digital form and digitalization is the application or adoption of digital technologies (OECD 2020).

The rapid acceleration of the digital revolution was spurred by the highly connected global economy with little to no blockades, along with the demanding nature of the COVID-19 pandemic and its stringent health regulations. Being forced into confinement through social distancing while having to keep economic activities going requires a fast-acting solution—the digital solution. At its center is the usage of digital technologies to increase connectivity (OECD 2015), along with its rapid diffusion into the economy, creating prospects of digitally fueled economic growth (Bukht and Heeks 2017). Having become an undeniable source of national income and a stimulus for further growth and development, understanding the digital economy and sharing economy is necessary to map the digital solution. The definition of the digital economy (DE) is a prevalent issue within economics, as the saturation of digitalization can be seen spread across the entirety of the economy, which then poses the problem of inclusion. Early definitions of the DE showed no direct or fixed definition, as explained by Tapscott (2014) and Lane (1999), where the DE is defined as the networking and communication of machines in parallel with the networking of

humans through technology resulting in the flow of information and technology. More recent definitions remain general, such as the socioeconomic functions and activities carried out by means of digital technology which encompasses physical infrastructures and the functionality they provide (Dahlman et al. 2016). The lack of a fixed definition has created a market void where there is no fixed framework of measurement of the DE, and all measurements are subject to domestic conditions and data availability. Where there is a literary gap, theorists often strive to fill it, and in the case of the DE, the works of Bukht and Heeks (2017) have often been cited to observe the performance or progress of the DE within a specific locality based on the concept of the segmented digital economy. The concept posits three stages of the DE, which start with the core digital sector, digital economy, and finally, the digitalized economy. Each stage requires specific milestones of the DE to be achieved before the domestic market can move on to the next stage.

The sharing economy, however, seems to have taken on more tangible definitions on its bounds and parameters. A more common consensus on the sharing economy (SE) has been understood as an economic model driven by the peer-to-peer exchange of goods and services supported by the factor of digitalization seen in the form of digital platforms (Dabbous and Tarhini 2021; Daunorienė et al. 2015; Hernandez-Carrion 2021; Mi and Coffman 2019; Yeganeh 2021). Similarly, Botsman (2015) referred to the sharing economy as the valuation of unused assets through a decentralized network as a departure from the traditional economic system, which is enabled by capitalization and pricing. The relevance of the modern sharing economy system is seen in the role which digitalization and technological innovation and adoption play. The recurring role of digitalization can be observed in the work of (Liu and Chen 2020), where the sharing economy is said to provide an avenue for value and utility creation from underused assets, with digital platforms as a medium.

Establishing the basis of the digital economy and sharing economy then brings the question of the role these segments play in the rhetoric of sustainable development. The increasing focus on sustainable development, especially amidst disrupted supply chains and economic and political warfare from Ukraine–Russia war and the US–China trade war, calls for the eradication of dependency on traditional systems based on nonrenewable energy and inefficient governance of the economy. The growth of socioeconomic and political unrest in recent times has impacted the flow of energy in the global economy. Taking the case of Germany, for example, the impact of the Russian invasion of Ukraine has the German government scrambling for alternative energy sources which, prior to the invasion, was heavily reliant on Russia's natural gas, comprising 55% of the total gas consumed in the country (Oltermann 2022). The role of digitalization in the usage of energy lies in the dissemination of energy, where digitalization has a stake hold in the supply value chain. Additional systemic diffusion of digitalization paired with a somewhat monopolized market creates a heavy reliance on market powerhouses.

Recent trends and expectations of the flow of natural gas see the gas pipelines policy from Russia to remain tight as shown in Figure 1, with expectations of further reduction in supplies to Europe International Energy Agency (2022b). This news comes as the Federation retaliates against the sanctions imposed on them as a response to the Ukraine invasion, which is a sour note as the European continent makes its way into the winter season, as well as for any other external markets which rely on the supply of liquefied natural gas (LNG). The most visible impact of economic impact of this invasion is seen in the scrambling of resources to procure LNG through various means, causing the tightening of the market and demand destruction, leading to the price crisis that is presently being felt across the entire economy.

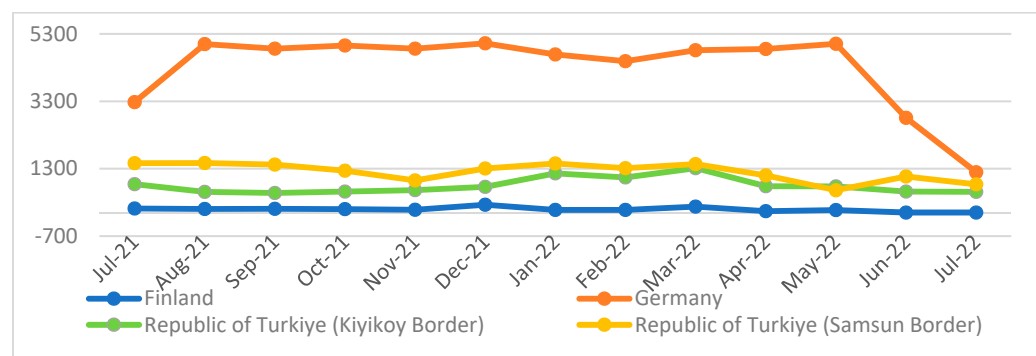

**Figure 1.** Russia gas outflow to selected countries (physical flows). Source: International Energy Agency.

　　Figures 2 and 3 show the trend of energy usage in the global arena up to the year 2020, which had been on a downtrend well before the recent geopolitical constraints. These figures indicate a deceleration in global energy usage while, simultaneously, the consumption of renewable energy experienced an increase, in which the International Energy Agency (2022a) reported at least three percent year-on-year growth from the year 2020 to 2021 for selected technologies, as seen in Figure 3. The deceleration of nonrenewable energy consumption comes in light of several factors, such as the increase in renewable primary energy demand and a growing capacity of solar and wind energy (bp 2022), which seems to indicate a growing market for sustainable energy production and consumption. This is also supplemented by the increasing utilization of existing and emerging technologies, digital technologies included, which could potentially halve the volume of global emissions, given the flow of necessary investments (OECD 2022).

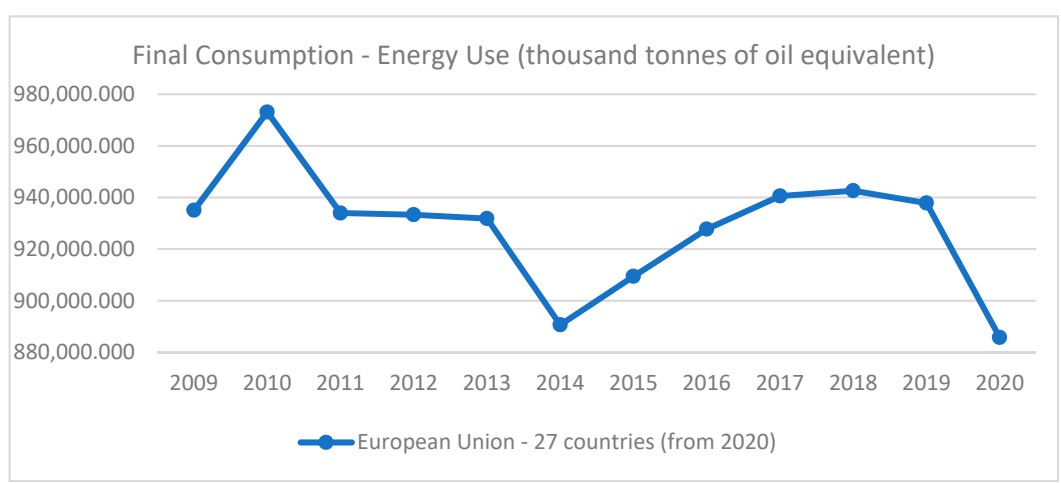

**Figure 2.** Final consumption–energy use (thousand tons of oil equivalent). Source: Eurostat.

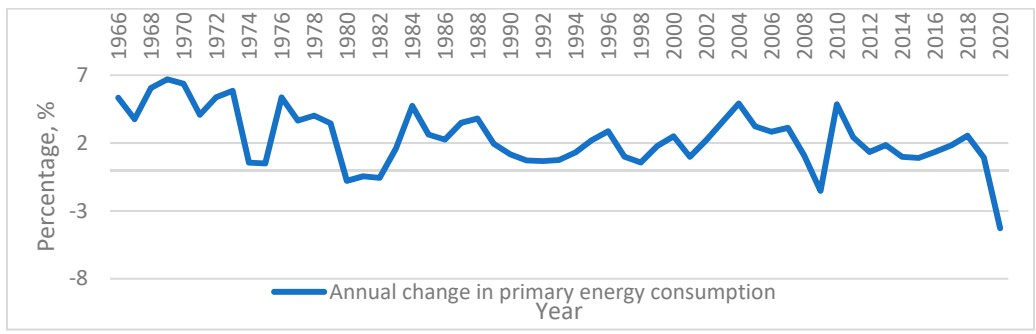

**Figure 3.** Global annual change in primary energy consumption. Source: Our World In Data.

The case of Malaysia is no different on the frontier of decelerating energy consumption, as can be seen in Figure 4. The trend of energy intensity, which measures the level of energy necessary for the production of one unit of output relative to the GDP, has been on a downtrend in recent years, indicating a decreasing reliance on energy or the increase in energy consumption efficiency for both industrial and household consumption. This downtrend holds large significance in the consumption of energy based on sources from both the global and national level. As coal and natural gases hold the biggest shares in energy consumption and production, the downtrend in energy consumption is majorly contributory to the reduction in reliance on these two nonrenewable sources and an indication of a gradual shift towards other energy sources for commercial and household consumption. In the discourse of sustainability, these data solidify the growing attention this subject matter holds and the need for further investigation into it, as well as the further growth in efficiency when paired with the current economic trend—the digital economy. All the data above take into account the morphing economic environment, where the digitalization process is reflected in terms of promoting production efficiency.

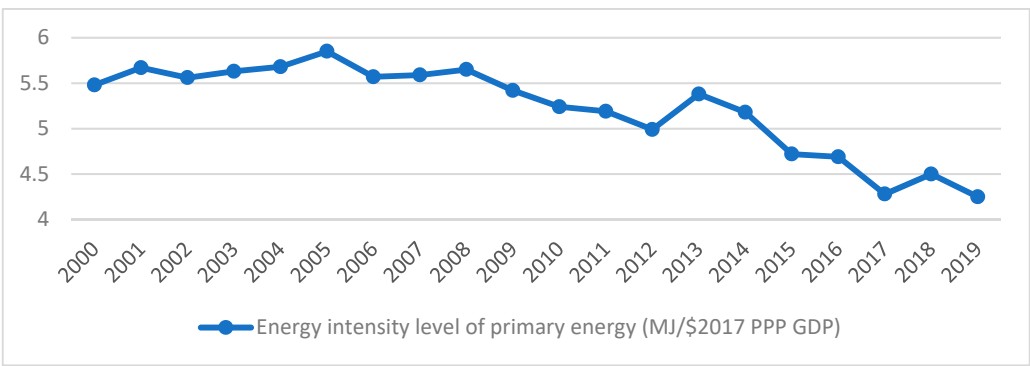

**Figure 4.** Malaysia energy intensity level of primary energy (MJ/USD 2017 PPP GDP). Source: International Energy Agency.

The case of Malaysia stands out from its neighboring countries and on a global level where, as a developing country, the promise and prospects of a faster rate of digital transformation are prevalent, with additional factors such as ease of market access, cheaper skilled labor and endowment of resources to sustain Foreign Direct Investments (FDIs) in the technological space. Injections in this space, paired with the right set of policy regulations, enables further growth in digital diffusion and increased dedicated efforts towards sustainability goals.

This paper dives into the ecosystem parameters of the DE, SE and sustainable development based on past research and attempts to establish a relationship among these segments of the economy. The objectives of this paper are outlined as follows:

**RO1.** To evaluate the current trend of publication of digitalization and sustainable development based on the performance of publication and other bibliometric analysis.

**RO2.** To analyze the content and results of the selected studies. For the purpose of granularity, this research objective is broken into subobjectives, which are as follows:

- To categorize the main streams of research from existing studies on digitalization and sustainable development.
- To identify the linkage of the digital economy and sharing economy.
- To investigate the role of digitalization in sustainable development.

**RO3.** To detect research gaps and limitations and provide recommendations for future research directions.

This paper utilizes a review analysis for exploration of the relationship between the digital economy and sharing economy, as well as the relationship between digitalization and sustainable development. As covered in the following sections, the relationship

shared between the digital economy and sharing economy is not explicitly explored and determined, whereas the relationship between digitalization and sustainable development is observed to take on several forms, such as positive, negative and indirect relationships, specifically in Stream 2 of the content analysis of this paper.

## 2. Review Methodology

### 2.1. Methodology and Study Framework

The study utilizes several methods in the analysis stage, such as bibliometric analysis, content analysis and systematic literature review (SLR), to provide a quantified qualitative review on the frontier of digitalization and sustainable development. Bibliometric analysis is the attempt to quantitatively assess the academic quality of journals or authors by statistical methods, such as citation rates (Ahamer and Kumpfmüller 2014; Azevedo et al. 2022). This method of analysis is useful in extracting publication trends on the studied research areas, which in this case are the digital economy, sharing economy and sustainable development, from the bibliographic information of the past literature. The study takes the review further through content analysis and the SLR method, which enables the literature and its contents to be systematically categorized, creating an ease in analyzing any gaps in the existing body of knowledge and offers recommendations for future research in a similar space and field (Paul and Criado 2020). The SLR allows for the most relevant literature to be chosen to be reviewed in this paper, ensuring accuracy of input. Adopting a framework derived from the PRISMA guideline proposed by Moher et al. (2009), the study organized its search and analysis of literature into four main steps, which are identification, screening, eligibility assessment and identification of the findings. The cumulative impact of utilizing such a holistic methodology and study framework is transparency in the entire review process of the study. The cumulative effect of the applied methodology addresses the gap of one-sided analysis, where both publication trends as well as systematic and precise content analysis are conducted. This then provides transparency in the entirety of the review process applied.

### 2.2. Data Source and Collection

This study obtained its set of past literature from the Web of Science and ProQuest databases based on stringent selection sourcing and collection criteria. The search for the literature for review in this study was conducted on two scholarly databases, which are ProQuest and Web of Science, in order to optimize the volume and thoroughness of the search process. The search strings are based on the keywords "Sharing Economy" and "Digital Economy". Following that was the application of the relevant key filters, as seen in Table 1, to eliminate unwanted or irrelevant records. The chosen articles extracted from the respective databases were further filtered based on abstract, content reading and relevance to this study, producing a total of 57 articles. The sample used in the bibliometric analysis, however, does not encompass the combination of all the searches conducted as part of the systematic literature and instead uses the search sample of the digital economy, which is n = 20,896. The search criterion and method for article selection can be seen in Table 1 and Figure 5. The data search and collection process takes on a funnelling method where a broad search is conducted, and specific filters are applied to the search based on the search criteria, which cover the inclusion and exclusion criteria for the articles to be reviewed.

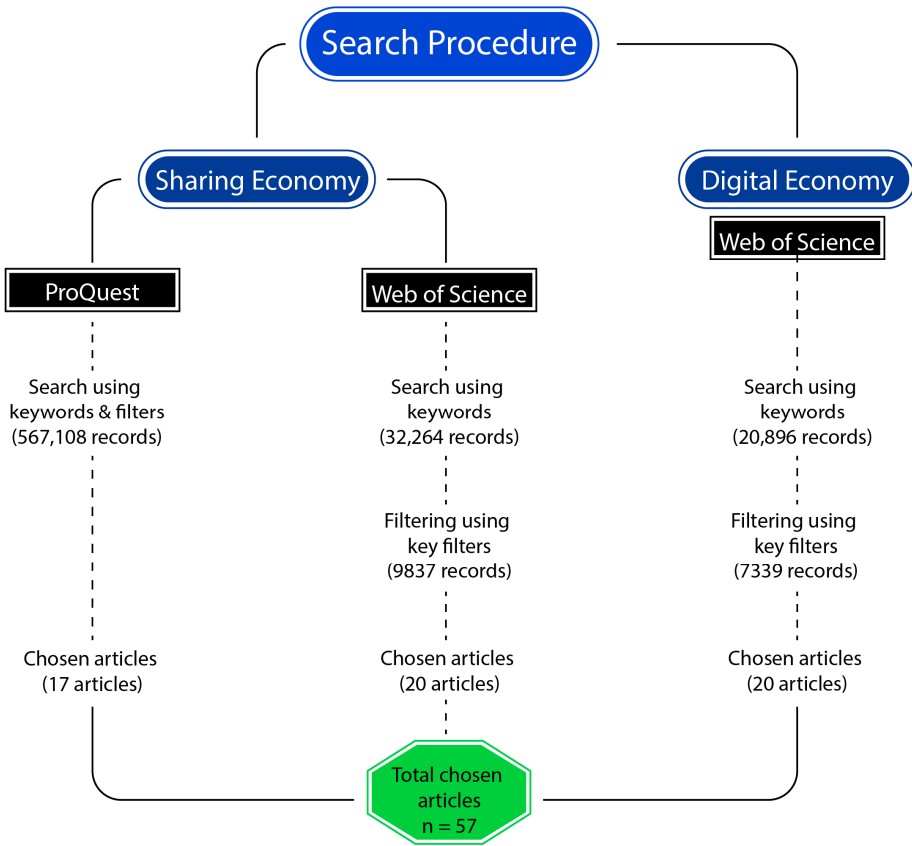

**Figure 5.** Selection procedure overview.

**Table 1.** Search criteria and method for article selection from ProQuest and Web of Science databases.

| Search Methods | ProQuest Database | Description |
|---|---|---|
| Searching for articles using keywords and filters | | The keywords "Sharing Economy" were applied along with following filters: |
| Filtering relevant records using key filters | 567,108 records | The articles are to be "Dissertation and Theses", "Scholarly Journals", "Conference Papers and Proceedings", "Books", "English" and "Full-Text". |
| Chosen articles based on criteria | 17 articles | Articles chosen are read through and filtered directly from the database and picked with factors in relation to sharing economy and sustainable development. |
| **Search Methods** | **Web of Science Atabase** | **Description** |
| Searching for articles using keywords | 32,264 records | The keywords "Sharing Economy" were applied. |
| Filtering relevant records using key filters | 9837 records | The articles are to be "Articles", "Review Articles", "Books", "Book Chapters", "Open Access" and "English". |
| Chosen articles based on criteria | 20 articles | Articles chosen are read through and filtered directly from the databases and picked with factors in relation to sharing economy and sustainable development. |
| **Search Methods** | **Web of Science Database** | **Description** |
| Searching for articles using keywords | 20,896 records | The keywords "Digital Economy" were applied. |
| Filtering relevant records using key filters | 7339 records | The articles are to be "Articles", "Review Articles", "Books", "Book Chapters", "Proceedings Papers", "Open Access" and "English". |

**Table 1.** *Cont.*

| Search Methods | ProQuest Database | Description |
|---|---|---|
| Chosen articles based on criteria | 20 articles | Articles chosen are read through and filtered directly from the databases and picked with factors in relation to sharing economy and sustainable development. |
| **Total chosen articles from ProQuest and Web of Science database** | **57 articles** | |

The umbrella term "Digital Economy" was put through a search stream in only one database due to the common and consensual findings on the digital economy. The high volume of research conducted on the digital economy has led to a deeply explored horizon with respect to specific research areas. The commonality in findings is not seen in terms of the sharing economy, which motivated the utilization of two databases to expand the search horizon and obtain a larger sample pool of articles to back up the trend analysis and content analysis explored in later sections.

Given that the paper is intended as a study on the DE and SE from a more macroanalytical overview, the first stage of the data collection applied the search keywords "Sharing Economy" and "Digital Economy" in its preliminary search stage. These keywords were applied on both of the aforementioned literature databases to ensure consistency in the search procedure. As a result of this stage of data curation, the number of results was overwhelming, which incited the need for the keywords to be paired with dedicated filters for the elimination of irrelevant search results. The filters applied in the Web of Science and ProQuest databases differed based on the database's interface and search options. The filters and filtering process are simplified, as shown in Figure 6 and Table 1, where the number of articles is whittled down from a population size to a sample size of 57 articles to be reviewed.

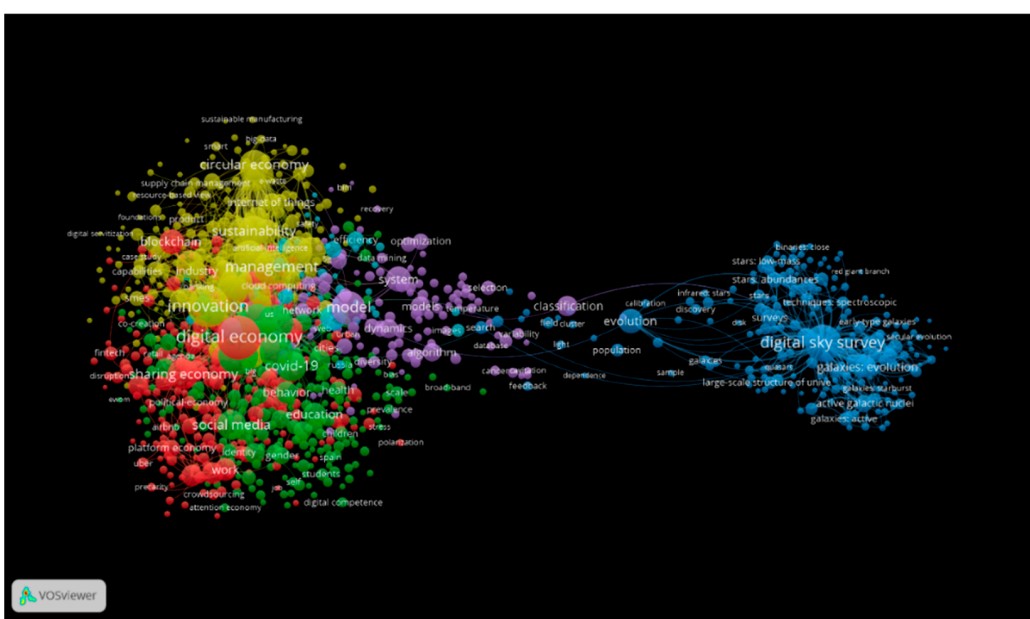

**Figure 6.** Co-occurrence network visualization (n = 20,896). Source: VosViewer.

Following Fink's (2019) definition of a rigorous stand-alone literature review, it needs to contain four main aspects, which are systematic in following a methodological approach, explicit in revealing the utilized procedures, comprehensive in its inclusion of relevant material and reproducible by those who adopt the same approach in a similar research area. This motivates the use of the systematic literature review to ensure that the four main aspects as covered by Fink (2019) are fulfilled and a rigorous literature review paper is

produced. A search guide is then adopted from the work of Alshater et al. (2022), where the steps of data sourcing and collection are simplified into four steps. As covered above, the first step is the keyword identification for data curation, followed by the actual search. While there may be vast development and an increasingly broad horizon on the studies of digitalization in the economy, this study focuses on the broad nature of the digital economy and sharing economy, especially in regard to sustainable development. These act as broad keywords which encompass the general definition of these research areas.

The second step is the data screening, which is essentially the filtering process in the funnel method. The articles in the search results from the previous step are thinned out using filters based on the search criteria shown in Table 1. The process applied on the ProQuest database differs from the Web of Science database process. The overwhelming search results from the keyword search required a direct application of filters simultaneously with the keywords to significantly whittle down the number of articles to go through the third step.

The third step involves the critical analysis of the entirety of the chosen articles from the prior step. The full text of the articles is reviewed based on the inclusion and exclusion criteria decided and set above. The full-text analysis conducted critically screened the full texts for research areas pertaining the digital economy, sharing economy and the relationship of these subject areas towards sustainable development, either ecologically or economic sustainability. The sum of the efforts in the first three steps produced a total of 57 chosen articles used for review in this study.

The final step is the categorization and summarization of the literature findings. The purpose of the categorization is twofold, where the first is for the simplification of the bibliometric analysis and the second for the ease of literarily clustering the full-text findings in the summarization portion of the study. The study analyzes the existing research gaps, limitations and research directions for future research, as well as the current research arena of the studied research areas through the trend analysis that is the bibliometric analysis.

### 3. Analysis

*3.1. Bibliometric Mapping*

The bibliometric analysis encompasses the analysis of a rough population of the digital economy literature search results, as well as the chosen sample of literature pertaining to the DE, SE and sustainable development. The study uses the DE literature search as a normative parent term for the aspect of digitalization. The flow of the bibliometric analysis begins with the literature population analysis, followed by a narrower scope of literature, which is the chosen literature to be reviewed in this paper. Based on the visualizations shown and explained below, the theoretical relationship within the literature space is fairly objectively scrutinized.

Figures 6 and 7 illustrate the co-occurrence of keywords among the literature of the DE population literature. It can be observed from the network visualization that the clusters of keywords with terms such as "Digital Economy", "Sharing Economy" and "Sustainability" are of significant prominence. The network visualization shows the linkages of issues and topics discussed within and among the scholarly articles. The interconnected clusters of keywords prove the relevance of the DE, SE and sustainability discussed within the same context and its applicability of reference to this study.

Zooming in on the relevant clusters from the Co-occurrence network visualization based on the n = 20,896 sample size, there are three main clusters of interest to this study, as mentioned earlier, which revolve around the digital economy, sharing economy and sustainability, as noted by the designated prominent colors of those clusters. While the general co-occurrence network visualization of Figure 6 shows the existence of these terminologies within the same environment with their respective degrees of interconnectedness, the purpose of zooming into each term or phrase of interest is to reveal the specific and closest relationships of it towards other terms. The digital technologically influenced economies are noted with a red accent in their cluster, whereas the aspect of sustainability is noted

with a yellow accent in its cluster, as seen in Figures 7–9. A prominent takeaway is the relationship among the terms, as well as the relationship between the terms and their surrounding terms. It can be observed that the SE and DE are often discussed in relation to sustainability and vice versa, based on the existence of the network and size of terms' bubbles in Figure 10. There is an additional cluster of interest which most prominently encompasses technology and the COVID-19 pandemic. The links of these areas are commonly seen as drivers of the rapid digital growth over the last few years, giving way to the development of the digital economy, sharing economy and sustainable development.

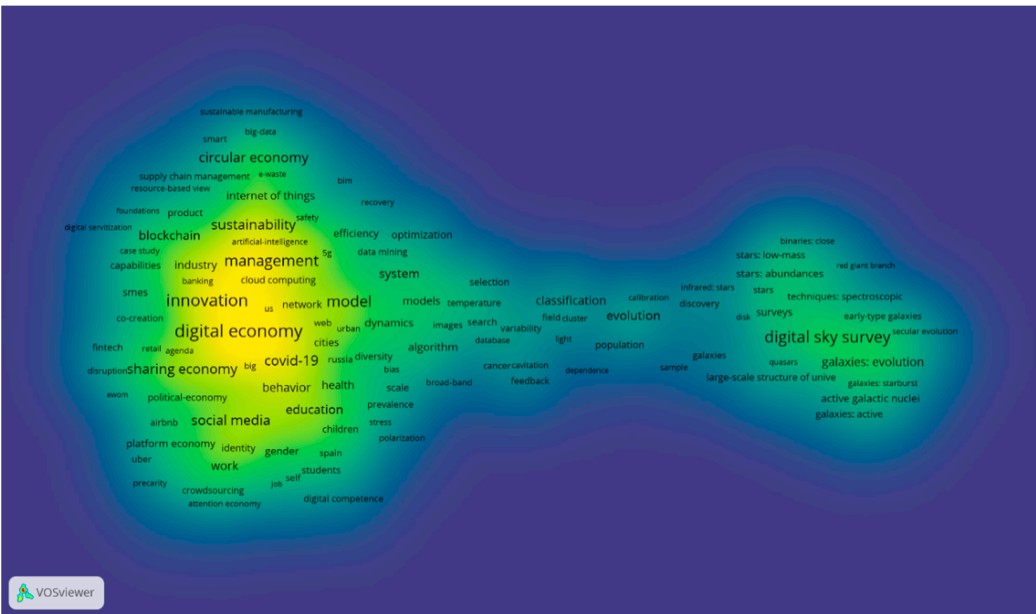

**Figure 7.** Co-occurrence density visualization (n = 20,896). Source: VosViewer.

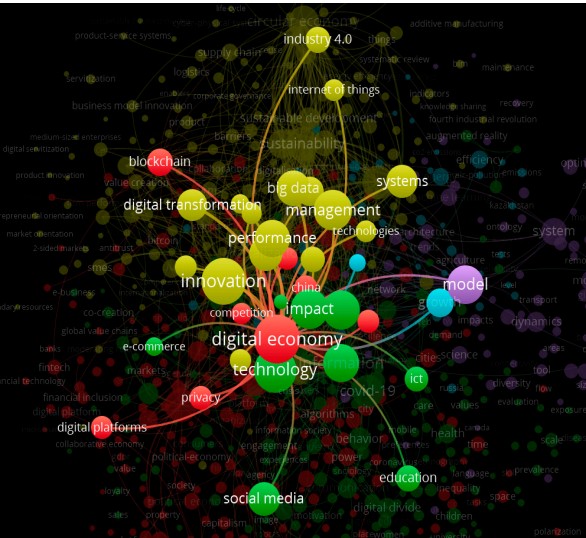

**Figure 8.** Digital economy cluster network visualization: n = 439 links; 1212 total link strength; and 369 occurrences.

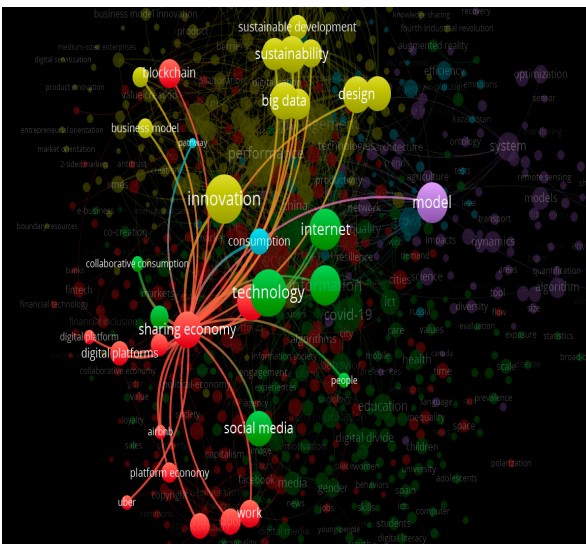

**Figure 9.** Sharing economy cluster network visualization: n = 339 links; 1006 total link strength; and 187 occurrences.

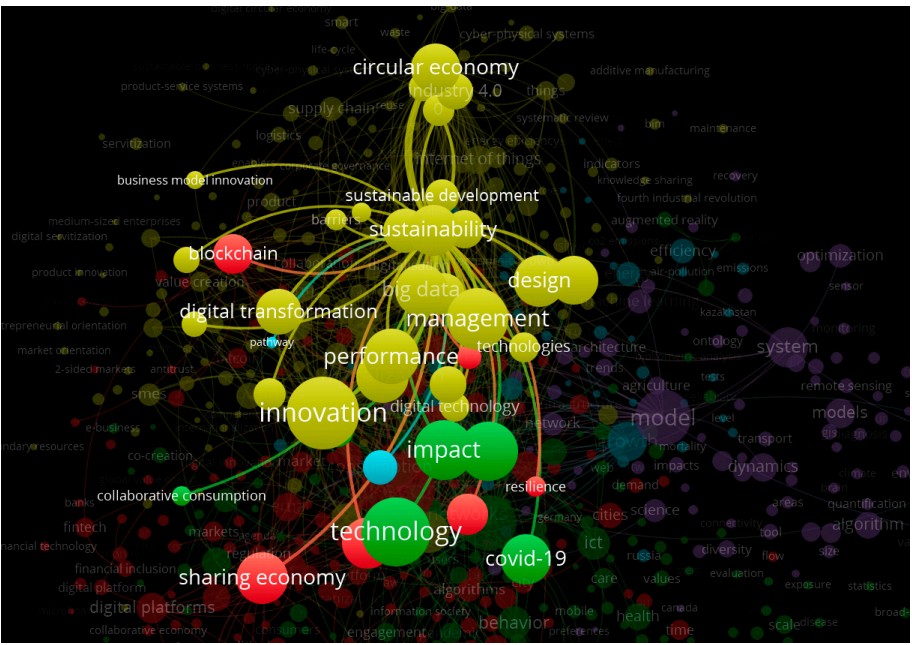

**Figure 10.** Sustainability cluster network visualization: n = 370 links; 1114 total link strength; and 182 occurrences.

Looking at the density visualization of the keyword co-occurrences, the weightage of the keywords, measured by the frequency of occurrence for each keyword, can be interpreted as a reference for the trend of research area focus. The larger the weightage of the keyword, or rather the higher the density of the keyword, the darker the hue and saturation of yellow surrounding the specific word or phrase. Similar to the network visualization, the population of the literature from the DE search sees a large focus on areas pertaining sustainability, management, sharing economy and circular economy, among other areas.

Another implication of these visualizations from the volume of keyword co-occurrences suggests a growing diversity in research areas, with the digital economy as the focus. The diversity of research areas is observed based on the nodes within the network visualizations, in which each node portrayed in network visualizations relevant to keywords and

recurring words shows a discussed item. The application of the bibliometric analysis is unique, as its application can cover several categories of analysis, including, but not limited to, co-authorship (Figure 11), bibliographic coupling (Figure 12) text (Figure 13) and keyword co-occurrence (Figure 14), For Figures 13 and 14 the focus was on The resulting nodes based on each of the categorical analyses reflect the extracted information from the sample of literature and its network strength in relation to each other. Thus, based on the bibliographic analysis conducted in this study, the range of links connected to the main interests of this paper—digital economy, sharing economy, and sustainability—is significant and diverse.

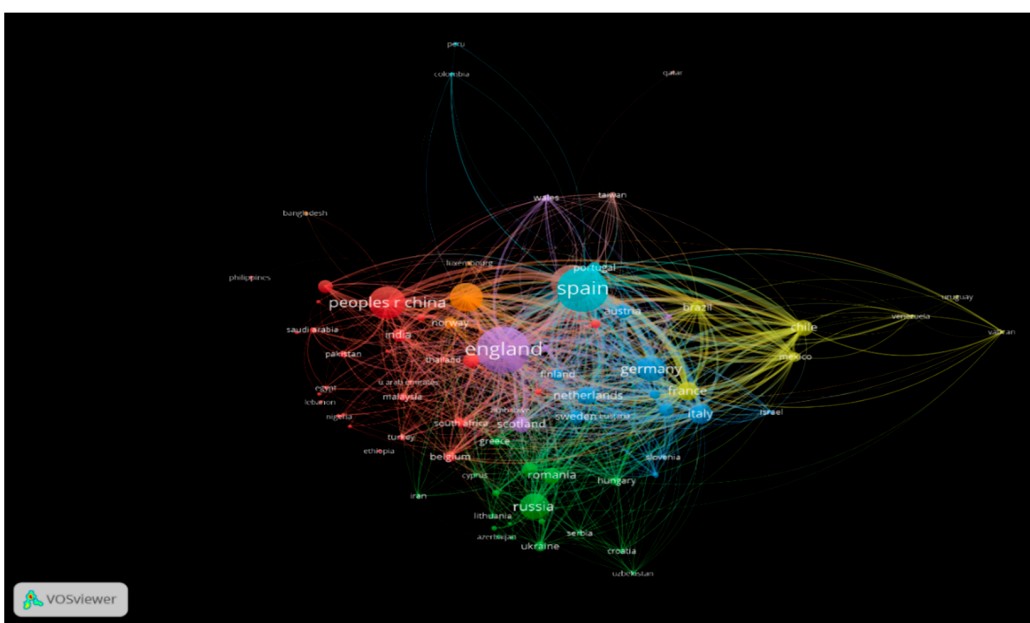

**Figure 11.** Co-authorship by country network visualization (n = 20,896). Source: VosViewer.

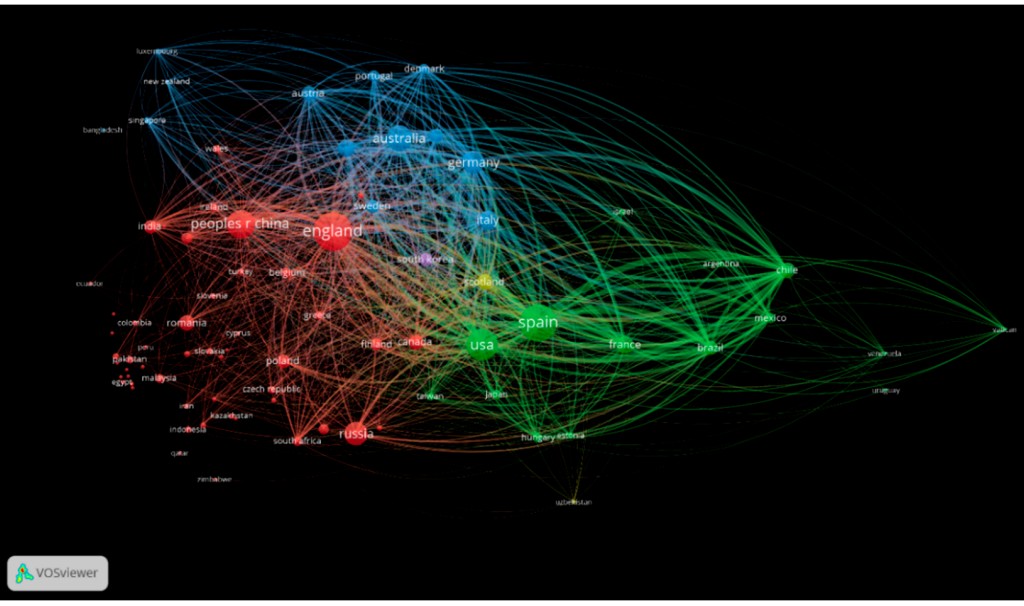

**Figure 12.** Bibliographic coupling network visualization (n = 20,896). Source: VosViewer.

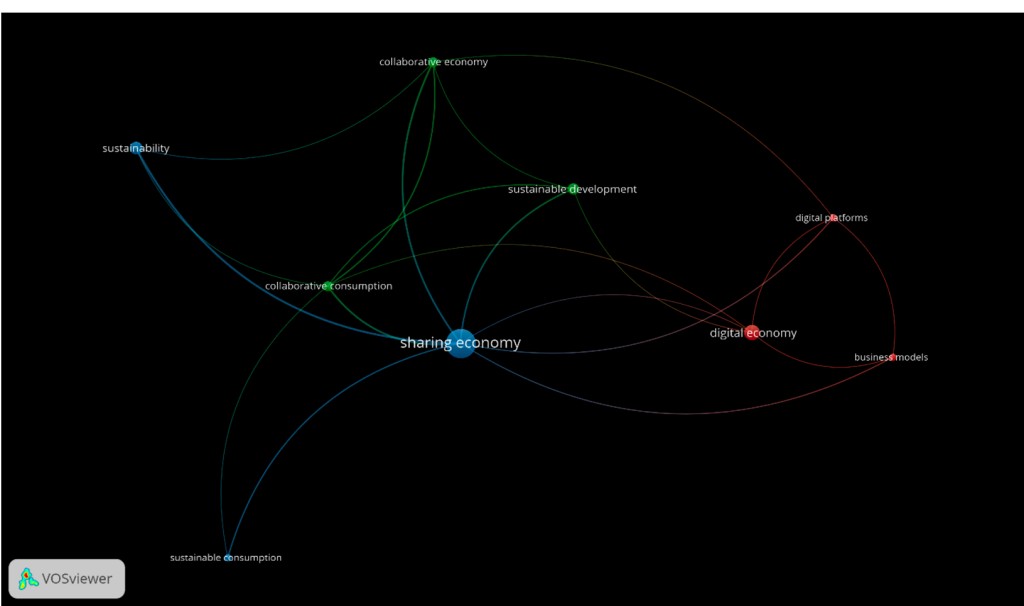

**Figure 13.** Keyword co-occurrence network visualization (n = 57). Source: VosViewer.

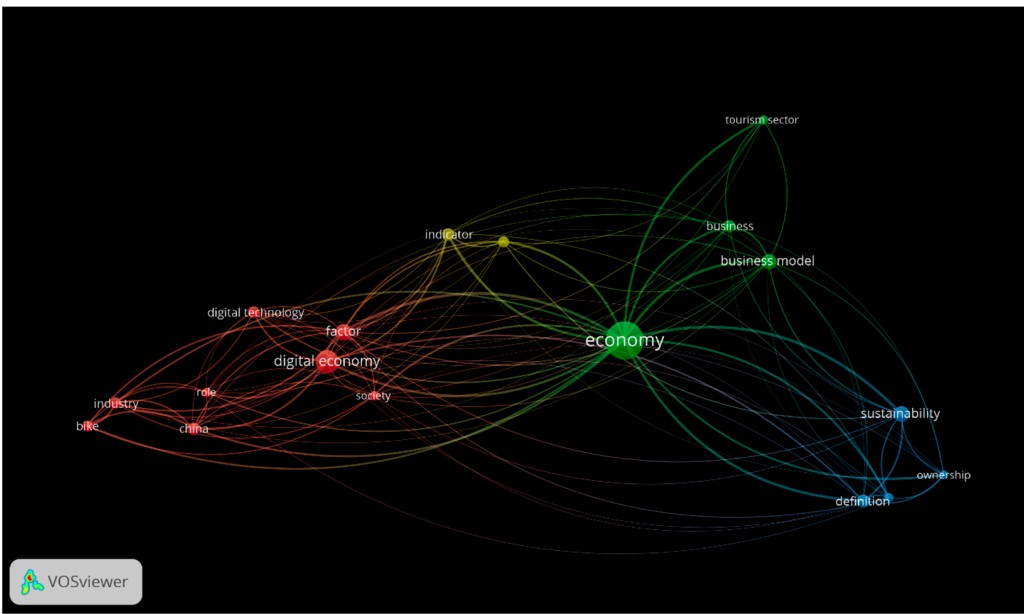

**Figure 14.** Text co-occurrence network visualization (n = 57). Source: VosViewer.

Figures 11 and 12 illustrate the co-authorship and bibliographic coupling as a factor of countries, respectively. The purpose of these illustrations is linked to the rising attention of the global economy towards the digital economy, giving rise to intellectual discussions to domestic and global participation in the digital economy. A common issue, as discussed by Ojanperä et al. (2019), is the degree of reliance of the national digital economy on the nation's wealth, as well as other influences of other macro- and microeconomic factors, which then brings into question the precision and accuracy of the measurement frameworks adopted by each country, which could wrongly depict the state of the domestic digital economy (Bukht and Heeks 2017). Thus, the state of literature surrounding the area of digitization has been experiencing an increase in exploration, both in terms of publication or readership.

The co-authorship and bibliographic coupling are representative of the collaborative aspect of the digital economy research environment and the interconnected nature of the

existing literature and body of knowledge. The large number of connective network lines and the strong network strength represented by the thickness of the lines in both the co-authorship and bibliographic coupling prove the global curiosity towards the subject of the digital economy and are a holistic global phenomenon demanding all perspectives to be considered. Another way to look at it would be the global collaboration in tackling the relatively new paradigm that is the digital economy with the help of the published literature from various countries, all the while adding more information to the current set of information on the digital economy.

*3.2. Chosen Articles' Bibliometric Mapping*

The microcontent analysis conducted in this section of the study is based on the sample of 57 chosen articles as funneled by the SLR in the earlier sections.

The set of chosen materials regarding the DE, SE and sustainable development has an obvious commonality in terms of keywords of the articles. Setting the number of occurrences to 10 to uncover the most used keywords, the network visualization of keyword co-occurrence in Figure 14 shows that the sharing economy, digital economy and matters of sustainability, among other keywords, are the most common within this field of research and specific search. The keywords here define the field, subfield or research area of the research paper, and the visualization based on the sample set of 57 chosen articles indicates that the bulk of the set share common research areas of interest to this study.

In comparison, in the co-occurrence of text data from the sample set of 57 chosen articles, three main clusters of terminology can be observed in Figure 13 which revolve around economy, digital economy and sustainability. A prominent difference between the network visualizations in Figures 12 and 13 is the lack of the term "Sharing Economy" in the text data co-occurrence. A possible explanation would be the lack of repetition in the discussion in the sharing economy, where while the sharing economy has its motivating benefits attracting consumers into the era of the digital platform economy, the advantage of this emerging market is made possible by digital infrastructure (Pouri and Hilty 2018), more commonly construed as an aspect of the digital economy.

Looking at the information of volume of publications from the chosen sample of literature to be reviewed in this study, the trend of publication can be observed in Figure 15. The earliest publication among the stringently chosen articles for the purposes of the literature mapping is seen to be 2015. With a gradual and fluctuating increase in publications from the year 2015 to 2018, the volume of publications with respect to the digital and sharing economy surged significantly starting in the year 2018. While 2019 saw a slight decrease as compared with the previous year, it still saw a significant jump in volume of research within the scrutinized fields. The drastic increase could be interpreted as rising awareness of and attention to the digitalization of the global economy, as well as any subsets of research areas relevant to these new forms of economy. This spotlight shone on the matters of digitalization and sustainable development within a short time frame shows the severity and need for attention and response by the global and respective domestic economies. The publications in the year of 2022 are low, as this study was conducted in the first half of 2022, up to the month of June where, presently, the volume of literature on these subject areas is still growing. For the literature to reflect the current environment, it is held at the mercy of the release of data, which can often prove run against the urgency of publishing a study.

The bibliographic analysis conducted in this study is conducted within a paradigm in which it acts as a macroreflection of the trends in the literature environment of the chosen subject areas. The components of the bibliographic analysis use the text, keywords and other information of the study as the input for the trend analysis, as well as for the investigation of relationships of subject areas and their network connections to other subject areas. The use of this form of literature analysis enables information to be illustrated in clusters, where it creates a clearer picture of the internal and external ecosystem of the digital economy, sharing economy and sustainable development.

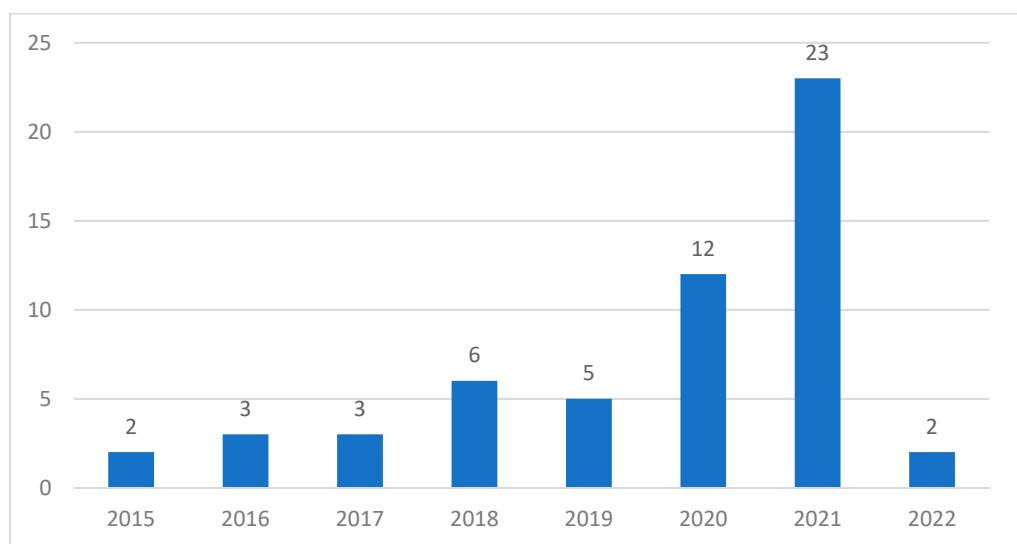

**Figure 15.** Number of publications by year (n = 57).

A study is not completed on the mere macrolevel, which brings the systematic literature review into the picture, which it is intended for a more microanalysis of the contents of the literature. Utilizing a strict selection process and selection criteria, the large sample of articles from the general search is stringently filtered through layers to end up with the most relevant and necessary literature as a point of reference for this study. The difference in the bibliographic analysis and systematic literature review may lie in the outcome of the analysis; however, the purpose remains the same, where they are both necessary simultaneously to properly map the literary ecosystem of digitalization and sustainable development, albeit focusing on macro- and microscales, respectively. The application of both methods simultaneously results in a funnel approach to the literature review, where analysis starts from a larger population, and the application of stringent selection criteria narrows the field and filters the literature from that sample. As seen in Figures 14 and 15, the sample is greatly reduced to 57 from the earlier 20,896 articles, which were derived from a general search of the digital economy.

Rather than utilizing the characteristics of the bibliometric analysis and systematic literature review separately, this study opted to apply both simultaneously to systematically analyze the existing body of knowledge on digitalization and sustainable development, in which the characteristics of either method may benefit off each other.

## 4. Content Analysis

This section is broken up into four streams which categorize the content of the articles systematically based on research area and themes. The categorization of the material content addresses RO2, which is the categorization of research areas into streams. The first stream also addresses the subobjectives of the second RO, which is the identification of linkages between the digital economy and sharing economy.

### 4.1. Stream1: Technological Infrastructure as the Driver of Growth

The increasing rate of development in the global economy has been in tangent with the development of the digital economy, stemming from an increased velocity of information transfer along with its availability to be very easily integrated across markets, sectors and industries (Kruljac 2021), producing a significant positive impact on total factor productivity (Tian and Liu 2021). This change in the path towards an information-reliant economy has influenced the need and development of ICT-supporting infrastructure, creating a socioeconomic environment revolving around the adoption of said digital infrastructure and its consequences (Baranov 2022). The socioeconomic benefit from the digital economy is the most significant in the sense of access for both the producers and consumers, consequently

influencing the patterns of supply and demand with employment, even in the existence of a polarized society in terms of gender inequality, urban–rural disparity or even income gaps (Wysokińska 2021).

While the benefits of the digital economy have been well discussed, especially since the dawn of the Fourth Industrial Revolution, it too poses a set of problems that are yet to be solved, mainly in terms of its definition, as it is subject to the fluidity of domestic parameters. At the present state of the technological revolution and the adoption of high technological systems, such as Big Data and Internet of Things (IoT), many economists and studies are citing the role of information technology and communication (ICT) infrastructure to represent the factor of digitalization, or at least part of it (Abendin and Duan 2021; Baranov 2022; Chinoracky and Corejova 2021; Jiang 2020; Kurniawati 2022; Li et al. 2021b; Su et al. 2021). There is a prevalence in differing quality and availability of indicators and methodologies across economies, leaving a gap in the digital economy ecosystem and giving rise to the need of a dynamic and agile framework (OECD 2020). The issue of exclusion is among the concerning matters with respect to the scope and measurement of the digital economy and can stem from both internal and external factors (Bukht and Heeks 2017).

Deriving from the works of Bukht and Heeks (2017), the scope and specificity of the definition of the digital economy has a direct relationship, creating the concept of the segmented digital economy, comprising the core digital sector, digital economy and digitalized economy. The growth in digital platforms as the next step of technological adoption in the economy indicates that the economy has made a transition into a narrower definition of the digital economy, participating in the sharing economy.

One of the main concerns of this study is on the shared relationship between the sharing economy and the digital economy. The definition of the sharing economy, as proposed by Botsman (2015), is the valuation of unused assets through a decentralized network as a departure from the traditional economic system powered by capitalization and pricing. Looking further into the literature of the sharing economy, the definition of this sector of the economy has taken on a more tangible framework to go along with respective variations according to economies, which generally work along the lines of the collaborative exchange of goods and services driven by platforms (Boar et al. 2020; Daunorienė et al. 2015; Hernandez-Carrion 2021; Trabucchi et al. 2019; Yeganeh 2021).

Another common point, essentially in the context of the modern sharing economy, is the recurring role of digitalization as an enabler of the contemporary sharing economy. As mentioned in the work of Liu and Chen (2020), the sharing economy gives an avenue to create value and utility from underused assets, with digital platforms as a medium. This is also supported by Daunorienė et al. (2015), Dabbous and Tarhini (2021) and Bai and Velamuri (2021), who argue that the role of peer-to-peer systems within the sharing economy is equated with the usage of digital networks to optimize the use of resources for lower cost and at higher velocity of trade, complemented by larger access and use of data and information. The involvement of technology as part of the sharing economy operations was initially seen by the part played by telecommunications in connecting members of the economy, be it business-to-business or peer-to-peer, simplifying the sharing economy model into service providers, users and digital platforms (Šiuškaitė et al. 2019). While there are socioeconomic disparities, along with human nature, which would normally inhibit such a market as the sharing economy, the adoption and development of digital platforms act as a governance system to coordinate the market and its activities (Bai and Velamuri 2021). With the existing definitions of the digital economy, however, they may vary from one another; they share the core role of technological adoption, particularly digital infrastructure across the entire economy. Thus, evidence from the sharing economy literature points out the necessity of digital platforms for the contemporary sharing economy to function, creating a linkage between the digital and sharing economy.

### 4.2. Stream 2: Digitalization on Ecological Sustainability

Taking on a different tangent in the literature review, the rise of discussion in the context of digitalization and sustainable development is growing in prevalence. The basis of the sustainable development discussion rose to the spotlight when the United Nations (2015) released the 2030 Agenda for Sustainable Development, along with the 17 Sustainable Development Goals (SDGs). In the context of digitalization and the sharing and digital economy, sustainable development is commonly discussed in two areas of study, which are sustainable growth of the economy and environmental concerns, as well as the role digitalization plays in it.

Looking into the environmental concern, there is a juxtaposition of digitalization and environmental sustainability, in the sense that to meet the environmental standards of some countries, all parameters of a product's lifecycle see a transaction cost increase (Vishnevsky et al. 2021). They also argue that ecosystem disruptions from the development and usage of digital innovation are also a real threat. However, the negative impact of digitalization is not as commonly seen as the positive impact. While the relationship of digitalization and environmental measure proxies such as carbon emissions may not be linear or direct, studies have shown that digital advancement aids in the reduction of carbon emissions and, in the case of the sharing economy, it provides alternatives to individual motor vehicle usage (Enochsson et al. 2021; Li et al. 2021a, 2021c). In the case of wastewater discharge, the study by Sun et al. (2022) discussed the influencing factor of the digitalization of the economy as a driver to develop the industrial structure as a means of efficiency and reducing polluting emissions into the environment. A consensus on the impact of digitalization towards environment sustainability has yet to be reached, as determining factors vary based on domestic conditions and varying factors; however, an interesting notion in the study by Curtis and Mont (2020) shows that the sharing economy and digitalization are not autonomously sustainable. A conscious effort is necessary in the implementation of Sharing Economy Business Models (SEBMs) to invoke their properties of environmental and market sustainability.

### 4.3. Stream 3: Digitalization and Sustainable Economic Development

On the other end of the spectrum is the discourse on sustainable economic development. The subject matter is a broad horizon in its entirety, and available research tackles certain parameters individually to understand how the components add up to a possible sustainable economic development. A prime element is the evolution of consumption patterns, as digitalization destroys the traditional systems. The rapid increase in digital adoption and participation in the sharing economy has driven an increase in reusage and maximizing utility from existing resources, drastically reducing the attention on nonrenewable resources in the consumption ecosystem (Pouri and Hilty 2018). Addressing the prevailing economic issue of resource scarcity, a change in consumption pattern towards reusage and utility maximization, among other factors, is driving the sustainability of the digitalized markets (Barbu et al. 2018). The topic of the contribution of digitalization, be it in the sharing economy or digital economy, covers the factors of social, economy, ecological and technological forces, which come off as the benefits of the development of digitalization (Šiuškaitė et al. 2019). The change in consumption patterns, as covered earlier, affects the socioeconomic environment, ecological factors and development of the digital markets to give a sense of long-term sustainability. On the supply side of digitalization, the factor of importance tends to surface in the discussions on sustainable economic development. The spillover effect of technological progress increases the capacity of business activities and production in a broader range and variety, all while reducing costs across all layers, creating development opportunities and avenues for more economic activities (Bonciu and Bâlgăr 2016). As opposed to unbacked opinions on the mechanization and digitalization across sectors removing job prospects from the markets, digitalization promotes employment absorption based on the development of industries and sectors, thus requiring more human capital to support the expansion and making it a factor to be included in

digitalization-related indexes, calculations and considerations (Chinoracky and Corejova 2021; Jiang 2020; Li et al. 2021c; Pouri and Hilty 2018; Su et al. 2021).

*4.4. Stream 4: Education and Awareness to Support Digitalization and Sustainability*

Just as in any other field, the role which knowledge and skills play in the development of a sector or industry is paramount. In mapping the platform economy infrastructure, Plewnia and Guenther (2017) cite the necessity of education as part of the digital infrastructure framework, noting the contribution of education in shaping and mapping digitalization. The place of education in the discussion on digitalization, especially with respect to sustainable development, is varied, but there is a concord on it representing the social aspect. Looking at the umbrella concept of the SDGs, a consensus on the positive significance of digitalization, especially the sharing economy, is prevalent in the literature, discussing outright the reduction in education disparities from the standpoint of human capital investment (Boar et al. 2020; Dabbous and Tarhini 2021; Perkumienė et al. 2021).

Another vector of knowledge possession is the social awareness of ecological concerns and significant positive economic possibilities. The proliferation of the digitalized economy can be attributed to environmental awareness (Curtis and Lehner 2019) and the awareness from the valuation of benefits derived from the participation in the digital economy and sharing economy driven by various factors, such as cost reduction and willingness to share (Bonciu and Bâlgăr 2016; Choi and Choi 2020; Lyaskovskaya and Khudyakova 2021).

*4.5. Gaps and Implications*

The study of the 57 chosen articles shows that there is a lack of material analyzing the relationship between the digital economy and sharing economy. The literature chosen for review in this paper shows that the studies into the digital economy and sharing economy are conducted separately. The content analysis of the sharing economy studies exposes the reliance of the sector on digital technology infrastructure, especially the contemporary sharing economy. Piecing the studies of the respective sectors together, the sharing economy appears to be a subset of the digital economy, which is essentially the digitalization of the economy. The implication of this review paper is to establish the definition of both the DE and SE and identify the type of relationship shared between the two.

The role of the DE and SE towards sustainable development is not prolifically studied, and where there are studies conducted, the results appear to be mixed. The results cover all possible bases, where studies have found digitalization to be positively and negatively contributory to sustainable development, and some have even found that it does not have an impact on sustainable development. The commonality in the research gap here is the lack of a fixed framework for the measurement of sustainable development, especially on the ecological front. The absence of a generalized framework creates further issues, such as the lack of efficient and accurate indicators, commonly swapped based on domestic circumstances.

Direct linkages are yet to be established between the digital economy and sharing economy, as well as between the digital economy and sharing economy towards sustainable development. The linkages are most commonly observed indirectly, as seen in past literature. As mentioned above, the past literature has found that the digital economy, proxied by the adoption of digital technology, is a driver and enabler of the contemporary sharing economy, which relies on digital infrastructure for the reutilization of assets. Mixed results are found in the case of digitalization and sustainable ecological development, but the digital economy and sharing economy are found to be contributors towards sustainable economic development.

## 5. Conclusions and Future Research Direction

Trends on the publications in the past few years have shown significant promise in the largely expanding areas of research with respect to digitalization and sustainable development. Higher volumes of publications and an increase in specific research areas

are contributing to the expanding horizon of literary space. At the core of it, focus is being placed on the digital economy and sharing economy, which are essentially economic business models of digitalization and also sustainable development—a prevalent ecological and economic issue globally. This study finds that there is an increasing focus on areas such as digital economy, sharing economy and sustainability, all driven by the growth and change in technology catalyzed by the recent pandemic.

The sharing economy exists with the support from the digital economy, or rather is a subset of the digital economy considering the reliance of the sharing economy on technological infrastructure, especially digital ones in the contemporary sharing economy context. Whereas the concept of the sharing economy has taken a more solid shape compared with the digital economy, it, however, remains obvious that the role the digital economy has in the sharing economy is that of an enabler, fortifying the earlier statement. This opens the door towards more microresearch in investigating the relationship between these research areas, especially in the space of the parameters of either, and how they affect each other in the face of shocks and booms. Future research direction prospects also outline the benefits of the construction of a generalized digitalization framework.

On the avenue of sustainability, there are mixed findings on the impact of digitalization in ecological sustainability. This study recommends the study of direct and indirect impact analysis on the factors of digitalization towards the mechanisms which define ecological sustainability. The abstract nature of digitalization paired with the broad range of factors encased in ecological sustainability demands specific and dedicated microstudies in this avenue. Dedicated indicators, including but not limited to the construction of indices, for both the measure of digitalization and ecological sustainability should receive attention in capturing a more accurate reflection of the current and desired future state.

The digitalization of sustainable economic development, however, proves to have clear positive implications, where the most prominent ones fall on the factor of efficiency. The adoption and utilization of digitalization in industries and across the economy allows for the efficient usage of resources, tackling one of the most basic economic issues: resource scarcity meeting unbounded demand. The digital factor within the economy creates the opportunity for the growth and development of businesses and sectors, increasing the capacity within these sectors matched with a lower and maximal capitalization of assets and resources. Parallelly, the growth of the sectors leads to the need for labor and human capital to match the speed of growth. A prospect for future research in this area is the stock take and review of parameters, indicators and analytical methodologies to ensure the most accurate reflection of the current economy. This creates the possibility of the study of the necessity of new indicators to measure the factor of technological change and its impact on the domestic or global economy.

Parallelly, there is call for the reduction, and eventually the eradication, of the reliance on traditional systems which are based on nonrenewable energy and the inefficient governance of it. The high reliance of the global and respective domestic economies paired with the high volatility of the geopolitical environment creates a void of vulnerability to internal and external shocks. The current geopolitical turmoil, for example, leaves one dominant power with control of the energy supply, while other countries are at the mercy of that power. With the growing realization of the importance of alternative renewable energy sources, there is a deceleration in nonrenewable energy consumption globally and a growth in renewable energy consumption. The increasing adoption of digitalization and its relevance to sustainable development has spurred mixed responses through studies on the relationship between the two. Concurrent growth and development in the space of digitalization and the growing importance of energy sustainability demand further studies on the impact of technological change on ecological sustainability.

**Author Contributions:** Conceptualization, O.A. and E.L.; methodology, O.A.; software, O.A.; validation, E.L.; formal analysis, O.A.; investigation, O.A.; resources, O.A. and E.L.; data curation, O.A.; writing—original draft preparation, O.A.; writing—review and editing, E.L.; visualization, O.A.;

supervision, E.L.; project administration, E.L. All authors have read and agreed to the published version of the manuscript.

**Funding:** Universiti Malaysia Sarawak: UNIMAS Graduate Grant F01/GRADUATES/2080/2021.

**Informed Consent Statement:** Not applicable.

**Data Availability Statement:** The data presented in this study are openly available in the Department of Statistics Malaysia, Bank Negara Malaysia, Eurostat Statistics, and International Energy Agency. All content analysis presented in this review paper is obtained from articles on Web of Science and ProQuest databases.

**Acknowledgments:** This research was financially supported by Universiti Malaysia Sarawak. This paper benefited from the valuable comments of the three referees and the editor of the journal on the earlier version of this paper.

**Conflicts of Interest:** The authors declare no conflict of interest.

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
