# Peer review of "Digital and Sharing Economy for Sustainable Development: A Bibliometric and Systematic Review"

_economies, doi:10.3390/economies11040105_

Round 1

Reviewer 1 Report

 I have gone through the manuscript entitled: “Digital and Sharing Economy for Sustainable Development: A Bibliometric and Systematic Review” and found it very interesting. However there is a slight modification required towards improving its readability:

Autor wrote “The sample used in the bibliometric analysis, however, does not include a combination of all searches conducted within the systematic literature”. It's worth adding why it doesn't include all combinations? What were the barriers to analysis?

Line 341 the volume of publications with respect to the digital and sharing economy surged significantly starting the year 2019. Are you sure this applies to 2019? Figure 16 shows an increase in 2021

Figure 1, 2, 3, it is worth citing the source of the data

Explain why Malaysia's example is worth citing. For example, is this country characterized by the greatest rate of change compared to other world countries? Is it therefore worthy of attention?

Figure 4 In the legend in yellow is bioenegry, and in purple is wind, but this is not in figure 4. It is worth changing the legend and removing this description if indeed Malesia does not use bioenergy and wind. I suggest just adding a sentence in the text explaining that these energy sources are not used.

Line 255  The digital technologically influenced economies are noted with a red accent in their cluster whereas the aspect of sustainability is noted with a yellow accent in its cluster as seen in Figures 9, 10, and 11.

There is no explanation for the green color which is also strongly visible on the chart. It is worth adding it for better readability and understanding of the text.

Author Response

Thanks for the comments. We responded accordingly as below.

  1. The research is broken down into several steps where first, a normative term of the digital economy was put through the analysis, and this was followed by a combination of the digital economy and sharing economy regarding sustainability. Following that was then an analysis of the sample pool of chosen articles from the search results. This enabled a micro analysis into each step of the way to identify trends based on the search. The sample also refers to the pool of chosen articles for review rather than all articles obtained in the search, creating a funnel effect of analysis of all articles to only the chosen articles.

  1. We have changed accordingly for Line 341 in the previous version which is now in line 382. Figures 1,2 and 3 are properly sourced (pages 2-3).

  1. Figure 4 has been amended properly. Also, refer to lines 127 – 133 on page 4. We deleted Figure 5 as it was redundant for the research.

  1. Line 255 was also amended with the provision of justification and explanation where necessary. The green cluster was first identified as a motivator of the digital economy, sharing economy and sustainability. However, as per the recommendation on improving readability, an explanation of the green cluster has been provided with linkage to the other clusters of interest as well (refer to line 295 onwards).

Reviewer 2 Report

This study intends to identify the relationship between the digital economy and sharing economy as well as their role in sustainable development. To achieve the objectives, this study use the bibliometric analysis and systematic literature review to organize and extract the contents of the selected literature.

Taking into account the stated goal, the contribution is processed in a very professional manner in terms of content. Individual parts follow each other, while the structure of the contribution remains clear.

Clear processing of the analysed data adds professionalism to the article. Figures are clear and are set in the right position.

Overall, the authors demonstrated thorough work with the sources, which is evidenced by the amount of literature used.

Based on the above, I have no suggestions for improvement.

Author Response

We are grateful for these kind comments from reviewer 2. We do take the liberty to amend the paper as in accordingly based on the other reviewer's suggestion. 

Reviewer 3 Report

Based on the content, the authors are trying to study trends in publication activity in the areas of digital economy, shared economy and sustainable development.

General comments:

While the topic of digital economy has been under study for a relatively long time, the topic of shared economy is innovative and insufficiently explored.

It is recommended to clarify the subject of the article. Based on the current topic, the article does not answer the question of whether the current state of the digital economy and the sharing economy is in the interest of sustainable development.

The annotation and conclusions of the article confuse the reader. Judging by the annotation, the article is research. Judging by the content of the article, methods and conclusions - the article is overview.

Authors need to determine whether their article is a survey or research. If this is a review article, the conclusions of the annotation should also address the trends in publication activity and content of these publications. The bibliometric and systematic review applied by the authors is not suitable for the topic of the article as research. It allowed the authors only to identify trends in publication activity, but did not address the essence of the phenomena of digital economy and shared economy. It is necessary to align the title of the article, its purpose, objectives.

It is necessary to show a gap in existing knowledge, formulate a hypothesis of research and consistently prove it.

Specific comments:

The annotation does not inspire to read the article. It is necessary to strengthen the novelty of the findings. In the article itself there are more interesting and new conclusions than those presented by the authors in the annotation. What new trends have you noticed in publications?

The findings about the ambiguous impact of the digital economy on the sharing economy with its network effects are not new. Don Tapscott also pointed out the dangers of network intelligence (1997). This warning, by the way, points to the volatility caused by the digital economy and the power of network intelligence.

Introduction. It is not clear why in the introduction the authors give an example of the Ukrainian-Russian conflict and data on energy supply. It is clear that any conflict or crisis leads to the destabilization of national or even global economies. This example is only about the overall sustainability of the economy and is not directly related to the digital economy, the economy of shared consumption, accompanied by network effects resulting from the use of digital platforms, which the authors rightly talk about.  In addition, the data given by the authors (July 2022, 2020)  have already become obsolete. Germany, as is known, has safely survived the winter. Inflation in Europe has been declining in recent months. It is either necessary to provide more justification for the link between energy use and the digital economy and the co-consumption economy, or to remove these theses.

In the introduction, the authors gave an overview of the current situation but did not show a gap in existing knowledge.

The purpose of the study is weak. What problems in the economy are affected by the article? The relationship between the digital economy, the sharing economy and economic sustainability is clear. If this link is questionable in the opinion of the authors, please provide more examples of publications (over the past five years) that have reached this conclusion.

The introduction should clearly articulate the research hypothesis, purpose and objectives.

Review methodology

Lines 156-157 - The authors point to the cumulative effect of the applied methodology on the transparency of the review process. The article does not solve the problem of increasing the transparency of the review process. The methodology needs to be not just described, but proved that this method is the best way to achieve the goal and solve the problems.

2.2 Data Source and Collection. Please specify in this section the period of selection of articles in ProQuest and Web of Science. Table 1 indicates that the keyword "digital economy" was searched only in the Web of Science database. Why the authors ignored ProQuest database, although they used this database for the keywords "sharing economy". It is not clear why the authors did not use Scopus. This could significantly expand the empirical base.

In Figure 5, the labels of the scales do not match the color of the columns. For example, lilac Wind, there are no purple bars on the chart. Similarly, the yellow Bioenergy is missing on the chart. If you choose to leave this picture, you must correct it.

The conclusions in section 5 are very general. It is necessary to summarize conclusions on trends in publications, which the authors have observed to give more specific recommendations for future studies of economic phenomena and problems. This would be important, especially for young scientists. It is not clear what the authors mean by «the construction a generalized digitalization framework».

Author Response

We are grateful and honoured to have Reviewer 3 for reading our paper. We take it seriously in revising the paper in accordance with the comments as below.

  1. The data and information in question have been updated and justified on their relevance to the creation of the background and formation of the story flow for the review paper as per the suggestions provided. The relevance of the Russia-Ukraine turmoil was provided to show the reliance of energy on a monopolising power and how technological sustainability is relevant in the context of this paper. A more in-depth explanation of the purpose of this study is provided as well to aid with the readability and relevance of this study. This can be found in lines 77 to 80.
  2. As mentioned in the paper, The research is broken down into several steps where first, a normative term of the digital economy was put through the analysis, and this was followed by a combination of the digital economy and sharing economy with regard to sustainability. Following that was then an analysis of the sample pool of chosen articles from the search results. This enabled a micro analysis into each step of the way to identify trends based on the search. The sample also refers to the pool of chosen articles for review rather than all articles obtained in the search, creating a funnel effect of analysis of all articles to only the chosen articles.
  3. We deleted Figure 5 as it was redundant for the research.
  4. We only adopted both WOS and ProQuest for the content analysis in the sample of n=57. Our focus database for this review paper was using WOS while ProQuest provide supplement support to our RQ, especially our microanalysis. Indeed, the SCOPUS database was another choice, but we do not pursue that. It is good for the next bibliometrics analysis to investigate and compare those from different databases. We also want to minimize the duplication between these two major databases.
  5. We have edited our conclusion to reflect the comments. As per the comments, a specific summary is included on the trend and content discovered through this review paper. This is paired with the more general summary given originally.

The readability of the paper is increased with the comments provide. 

Round 2

Reviewer 3 Report

I have not any comments.